# Generating Behavior-Driven Development (BDD) Artifacts

## Abstract

Behavior-Driven Development (BDD) specifies system behavior through scenarios, which can serve both as executable specifications and as automated test cases. In practice, BDD scenarios are created alongside large volumes of semi-structured or unstructured records (e.g., textual documentation, issue discussions, and informal feature descriptions). As a result, generating BDD scenarios and their accompanying records is often labor-intensive and error-prone. This paper investigates *bidirectional generation* between semi-structured or unstructured textual records and structured BDD scenarios using large language models. We conduct a comparison between (i) fine-tuning of CodeT5+ encoder-decoder models on aligned (record, scenario) and (scenario, record) pairs, and (ii) retrieval-augmented few-shot prompting with Meta-Llama-3.1-8B, CodeT5+ models, and DeepSeek-Coder. Experiments on a curated dataset of 2,100 aligned pairs show that generation quality is influenced by task direction and context management strategy, and that prefixing benefits tasks requiring strict structural output. Fine-tuned models achieve strong record-generation performance, with best BLEU/F1 of 0.9394/0.9549 (CodeT5p-770m, unprefixed-truncating) and best Exact Match of 0.8119 (CodeT5p-770m, prefixed-summarizing).

## Keywords

Behavior-Driven Development (BDD), Large Language Models

**ACM Reference Format:**
Anonymous Author(s). 2024. Generating Behavior-Driven Development (BDD) Artifacts. In *Proceedings of Make sure to enter the correct conference title from your rights confirmation email (Conference acronym 'XX)*. ACM, New York, NY, USA, 9 pages. https://doi.org/XXXXXXX.XXXXXXX

## 1 Introduction

Behavior-Driven Development (BDD) specifies system behavior using scenarios written in the *Given–When–Then* format, where *Given* defines the context, *When* specifies an action, and *Then* describes the expected outcome [12, 27, 35]. These scenarios capture expected behavior and can serve both as executable specifications and automated test cases [6, 7]. In addition to scenarios, projects contain many related textual artifacts that describe intended behavior in a less structured way. In this study, we call such artifacts *records*. A record is any semi-structured or unstructured text (e.g., REST API documentation, README files, .rst documentation) that relates to a BDD scenario.

Recent work has shown growing interest in using large language models (LLMs) to automate BDD-artifact generation, but most studies emphasize unidirectional generation (e.g., user stories to Gherkin), prompt-based methods, or industrial case studies [4, 13, 14, 22, 32]. Although some results are promising in specific settings [24], there is limited empirical evidence comparing model adaptation strategies under controlled conditions. In particular, the relationship between records and BDD scenarios remains under-explored, and prior work has not reported a direct comparison of fine-tuning and retrieval-augmented (RAG) prompting on the same task and dataset [18, 36]. Existing studies also rarely analyze the combined impact of task direction, instructional prefixing, and context management strategies, despite the prevalence of long real-world BDD artifacts (i.e., records, scenarios).

To address the gaps, we investigate bidirectional generation and compare (i) fine-tuning of CodeT5+ encoder–decoder models [34] on aligned (record, scenario) and (scenario, record) pairs with (ii) RAG few-shot prompting using CodeT5+ models [34], Meta-Llama-3.1-8B [26], and DeepSeek-Coder [16]. Records and scenarios are often longer than the 512-token limit supported by the encoder-decoder models. To handle this, we evaluate context-management strategies in the fine-tuning setup, including: truncation [10], abstractive summarization [33], and one-to-many chunking [21]. In contrast, for the RAG models, we did not apply specific length handling strategies like we did for fine-tuning, relying instead on the models' automatic truncation if the total prompt exceeded the context limit. Task direction was investigated across all models, but instructional prefixing was only tested in the fine-tuning experiments because the RAG setup already includes explicit task instructions by default. All approaches are evaluated on the same curated dataset using Exact Match [18], BLEU [29], and token-based F1 [30]. The evaluation is complemented by human evaluation and an LLM judge.

The results show that generation quality is influenced by task direction and context management strategy, and that instructional prefixing benefits tasks requiring strict structural output. Fine-tuned models achieve strong record-generation performance, with best BLEU/F1 of 0.9394/0.9549 (CodeT5p-770m, unprefixed-truncating) and best Exact Match of 0.8119 (CodeT5p-770m, summarizing). For scenario generation, the best BLEU/F1 is 0.6685/0.7697 (CodeT5p-770m, prefixed-truncating), while the best Exact Match is 0.0762 (CodeT5p-770m, summarizing), whereas RAG prompting achieves 0.00 Exact Match in all settings and BLEU scores below 0.35.

**Contributions.** (i) Created a dataset of 2,100 aligned pairs of BDD records and scenarios collected from 14 GitHub repositories. (ii) Conducted a comparison between fine-tuning and RAG few-shot prompting for bidirectional BDD artifact generation. (iii) Provided an analysis of task direction, instructional prefixing, and context management strategies for bidirectional BDD artifact generation.

The rest of this paper is organized as follows. Sections 2 and 3 cover related work and a running example. Section 4 describes our approach to BDD artifact generation using fine-tuning and prompt

engineering. Sections 5 and 6 present the experimental setup and results. Sections 7 and 8 discuss limitations and and future research.

## 2 Related Work

Recent work has investigated using LLMs to support BDD, where a common approach relies on prompt engineering. Karpurapu et al. [22] compare zero-shot and few-shot prompting across multiple LLMs for generating acceptance tests from user stories, reporting that GPT-based models achieve better performance with fewer syntax errors and higher validation accuracy. Similarly, Hiago et al. [13] study how model choice and prompt design affect the quality of generated Gherkin scenarios. Industrial case studies further show the feasibility of prompt-based generation for acceptance testing and test coverage improvement [14, 32].

Beyond general prompt engineering, several papers propose specialized tools and generation paradigms. Chang et al. introduce LLMScenario [8], which provides structured support for scenario generation. Other work frames LLMs as agentic systems for BDD-related tasks, where multiple agents collaborate to plan or execute testing workflows [28]. These tool- and agent-based approaches mainly focus on forward generation pipelines and typically evaluate performance within a single task direction.

LLM-based BDD generation has also been explored across diverse application domains. Zhou's thesis [36] surveys methods for automating BDD with LLMs. Domain-specific studies report promising results in contexts such as industrial security at Amazon [24], hardware design [11], and translating legal text into behavioral specifications [17]. Complementary research has examined downstream stages beyond artifact generation, including test execution [5]. While these works demonstrate breadth of applicability, they generally provide limited controlled comparisons of alternative adaptation strategies under consistent datasets and experimental setups. Finally, dataset availability remains a key bottleneck for systematic evaluation. To address this, Galloy et al. propose SelfBehave [15], which generates synthetic BDD data to facilitate training and benchmarking. However, synthetic datasets may not capture the length, noise, and formatting variability of real-world artifacts.

Unlike prior work that largely centers on unidirectional prompting or domain-specific case studies, this paper evaluates bidirectional generation between records and BDD scenarios on a curated dataset of real GitHub artifacts. It further provides a comparison between fine-tuning and RAG prompting, and examines practical factors that affect quality in real settings, including context handling, task direction, and instructional prefixing.

## 3 Running Example

In BDD, expected system behavior is expressed as *scenarios* written in Gherkin using the Given-When-Then structure. We use the example scenario shown in Figure 1 throughout the paper. It specifies a check for the *Insights Results Aggregator Mock* service to ensure the "content" endpoint returns the correct list of groups.

In this scenario, Given states the precondition (the system is in a default state), When states the action (requesting content and groups), and Then and And state the expected outcome (the response code is 200 and a specific list of groups is returned). Because the steps are explicit, the scenario can be automated and executed.

Figure 1 also shows an example record that corresponds to the scenario. This record references a feature file (e.g., insightsresults-aggregator-mock/content_info.feature) and lists high-level tasks like "Check if Insights Results Aggregator Mock service return correct list of groups." It also provides the necessary context for the test, such as the service hostname and port. A *Feature* is a standard BDD component that groups related scenarios under a common goal. Unlike a scenario, a record usually just states *what* should be verified but does not define the actual executable steps. The scenario defines an item from the record by translating that high-level statement into concrete steps: the precondition (Given), the action (When), and the expected outcome (Then). For instance, the task about checking the list of groups is implemented by a scenario that requests content and groups (When) and verifies that the response code is 200 with the correct table of data (Then). We can also see how the different approaches perform by looking at the inverse task of generating a record from a scenario.

## 4 Approach

This study investigates the generation of BDD scenarios from records and vice versa using LLMs. The approach is organized into two complementary parts: (1) fine-tuning of pre-trained models on curated (record, scenario) pairs for BDD scenario generation task and (scenario, record) pairs for record generation task, and (2) using prompt engineering with RAG few-shot learning for record and BDD scenario generation tasks. This two-part design enables a systematic comparison between parameter-updated learning and in-context reasoning for record and BDD scenario generation tasks.

### 4.1 Part I: Fine-Tuning

Figure 4 illustrates the overall workflow, which comprises two stages: (1) configuration and setup, and (2) preprocessing. During configuration, the model architecture and dataset including (record, scenario) or (scenario, record) pairs are selected, defining the learning direction. In this context an unprefixed input is the raw input text [scenario] or [record] without additional instructional prefix. To explicitly condition the model on the intended generation task, an instructional prefix was added to every training example in the prefixed setting. This prefix served as a task identifier and was prepended to the raw input text.

For the *record generation task*, each input followed a fixed instruction template of the form "generate record from feature: [scenario]". When long target outputs required chunking, the same input scenario was reused across multiple training instances, and the prefix was extended to indicate continuation. Specifically, subsequent chunks used prefixes such as "continue generating record (part i): [scenario]", with the part index (i) incremented sequentially according to the number of chunks. Similarly, for the *scenario generation task*, each input followed the template "generate test scenario: [record]". For chunked examples, continuation prefixes were applied in the same manner, resulting in inputs such as "continue generating test scenario (part i): [record]". This prefixing strategy provides explicit task and continuation signals, enabling the model to distinguish between generation directions and maintain coherence across chunked outputs. The preprocessing stage determines whether prefix-based or unprefixed inputs are used and applies one of several context management strategies, including

**Record**

Feature: Checking responses from Insights Results Aggregator Mock service: "content" endpoint
Background:
Given REST API service hostname is localhost
And REST API service port is 8080
And REST API service prefix is /api/insights-results-aggregator/v2
## ['insights-results-aggregator-mock/content_info.feature']
(https://github.com/RedHatInsights/insights-behavioral
-spec/blob/main/features/insights-results-aggregator-mock/content_info.feature)
* Check if Insights Results Aggregator Mock service return correct list of groups

Task 1: Scenario Gen.
Task 2: Record Gen.

**Scenario**

Scenario: Check if Insights Results Aggregator Mock service return correct list of groups
Given the system is in default state
When I request content and groups
Then The status code of the response is 200
And The status message of the response is "ok"
And I should retrieve empty content
And I should retrieve following list of groups
| Title | Description |
| Tags |
| Fault Tolerance | Load balancer issues, machine api and autoscaler issues, failover issues, nodes down, cluster api/cluster provider issues.
|fault_tolerance |
| Performance | High utilization, proposed tuned profiles, storage issues
|performance |
| Service Availability | Operator degraded, missing functionality due to misconfiguration or resource constraints.
|service_availability |
| Security | Issues related to certificates, user management, security groups, specific port usage, storage permissions, usage of kubeadmin account, exposed keys etc.
|security |

**Figure 1: Transforming a record into a scenario (top arrow) and regenerating the record from the scenario (bottom arrow)**

**Prompt RAG Output:**

```
Feature: Checking responses from Insights Results
Aggregator Mock service: "groups" endpoint

Background:
Given REST API service hostname is localhost
And REST API service port is 8080
And REST API service prefix is
/api/insights-results-aggregator/v2
## ['insights-results-aggregator-mock
/list_of_groups.feature']
(https://github.com/ RedHatInsights/insights-
behavioral-spec/blob/main/features
/insights-results-aggregator-mock
/list_of_groups.feature)

* Check if Insights Results Aggregator Mock service
return correct list of groups" }
```

**Figure 2: Obtained using llama-3.1-8b 3shot record**

**FineTuned Model Output:**

```
Feature: Checking responses from Insights Results
Aggregator Mock service: "content" endpoint

Background:
Given REST API service hostname is localhost
And REST API service port is 8080
And REST API service prefix is
/api/insights-results-aggregator/v2

## ['insights-results-aggregator-mock
/content_info.feature']
(https://github.com/RedHatInsights/insights
-behavioral-spec/blob/main/features/insights
-results-aggregator-mock/content_info.feature)

* Check if Insights Results Aggregator Mock service
return correct list of groups
```

**Figure 3: Example of record generation from feature file using fine tuned codet5p-770m model with instructional prefixing, one to many chunking, and summarizing strategy**

truncation [10], abstractive summarization [33], or one-to-many chunking combined with truncation or summarization [33]. Then, tokenization, training, and validation are performed on the processed dataset, resulting in model outputs that are subsequently used for evaluation.

*4.1.1 Context management strategies* A practical challenge in record and scenario generation from real-world artifacts is sequence length: both records and scenarios frequently exceed the 512-token context

limit imposed by encoder–decoder architectures such as CodeT5p. Context management strategies therefore play a critical role in determining model effectiveness. In this study, four commonly used strategies were investigated. The first strategy, *truncation* [10], establishes a baseline under strict length constraints. In this setting, both record and scenario texts were analyzed for sequence length,

**Figure 4: Fine-tuning workflow**

and any text exceeding the token limit was truncated to a safe boundary that accounts for special tokens. While effective at enforcing length limits, truncation may discard potentially relevant information from longer sequences. To reduce information loss, a second strategy based on *abstractive summarization* was used [33]. Instead of removing content, sequences exceeding the token limit were replaced with condensed summaries designed to preserve core semantic meaning while remaining within the allowable context window. Shorter sequences were retained unchanged.

Although truncation and summarization address length constraints, both approaches can result in information loss, particularly when the target output is long. To preserve the complete content of long outputs, a one-to-many chunking strategy was adopted. The implementation of this strategy first constrains the input text to the token limit, and then segments any long outputs into multiple smaller chunks. Each resulting output chunk is then paired with the same, single processed input, transforming one long example into multiple training instances. It is important to note that while this strategy results in a one-to-many mapping at the training data level, it maintains the fundamental one-to-one conceptual relationship between records and scenarios described in Section 5.

This chunking strategy was evaluated using two different context-management strategies. For truncation with one-to-many chunking, any input exceeding the token limit was first truncated. The single truncated input was then paired with each chunk of the corresponding long output. For summarization with one-to-many chunking, long inputs were condensed using an abstractive summarizer. This single summary was then paired with each chunk of the long output.

All context management strategies were evaluated across both task directions and with and without instructional prefixes, enabling a controlled analysis of how context handling, and task formulation affect fine-tuning performance.

### 4.2 Part II: Prompt Engineering

For the prompt engineering, we chose to focus on RAG because prior work shows it enables BDD artifact generation without model retraining [23, 24]. We split the same aligned pairs used for fine-tuning into two disjoint subsets: a *knowledge base* and a *validation set*. The knowledge base supports bidirectional retrieval through a vector database built from embeddings of the source-side text for

each direction. For scenario generation, records are embedded to retrieve similar records and their corresponding scenarios; for record generation, scenarios are embedded to retrieve similar scenarios and their corresponding records. In both cases, the full (record, scenario) pair is stored as metadata so the complete example can be inserted into the prompt.

For each validation input, we retrieve the top-$k$ most relevant records/scenarios and construct a prompt consisting of (1) a role definition, (2) retrieved input–output demonstrations, and (3) the task instruction. Figure 6 shows the prompt template for BDD scenario generation. We used a similar template for record generation.

## 5 Experimental Setup

This section describes the experimental setup used to study the BDD artifacts using fine-tuning and prompt engineering.

### Part I: Fine-Tuning

**Models** The fine-tuning experiments employed models from the CodeT5+ family [34]. We select CodeT5+ because it is a widely used encoder-decoder model family for code-related generation and translation tasks, and it provides a strong baseline for structured text-to-text generation. In addition, its 512-token context window imposes a realistic constraint for long BDD artifacts, enabling a controlled study of context management strategies (truncation, summarization, and chunking) for bidirectional generation. Two model variants were evaluated to study the effect of model scale on performance: `codet5p-220m` contains approximately 220 million parameters and serves as the smaller model, while `codet5p-770m`, with approximately 770 million parameters, represents a larger and more expressive configuration. Fine-tuning was performed using the Hugging Face `Trainer` API [20], enabling reproducible and standardized training across experimental conditions. Training hyperparameters were configured via `TrainingArguments`, including the number of epochs, batch size, and optimization settings. Table 1 provides a summary of the used parameters. Fine-tuned models were evaluated using three complementary metrics: Exact match [18], BLEU score [29], and F1 score [30].

**Computational Environment** Experiments involving the CodeT5p-220M model were executed on Kaggle using NVIDIA P100 GPUs. Experiments with the larger CodeT5p-770M model were performed on the Digital Alliance of Canada's Trillium compute cluster, utilizing four NVIDIA H100 SXM GPUs with 80 GB of memory each.

**Dataset** The dataset was constructed by mining public GitHub repositories using keyword searches (e.g., "Gherkin", "BDD") and file extension filtering for both `.feature` and `.rst` files. An initial set of 114 repositories was identified; however, many were excluded due to incomplete or missing documentation-scenario pairs. After filtering for coherence and completeness, 14 repositories were retained. Files that violated standard BDD conventions or contained nested or overlapping scenarios which were decomposed into parent scenarios. This process resulted in 2,100 paired instances of records and BDD scenarios. While these 2,100 pairs represent a unique one-to-one mapping at the conceptual level, some pairs are expanded into one-to-many training instances during the chunking phase to handle context length limits (see Section 4.1.1). The same dataset was used for both generation tasks. Figure 5 shows the steps taken to collect the data. Datasets were split into training

(80%) and validation (20%) subsets using fixed random seeds to ensure consistent splits across runs. Following preprocessing, all datasets were converted into a numerical format using a unified tokenization pipeline. The tokenizer corresponding to each target model (e.g., the RoBERTa-based tokenizer [25] used by CodeT5+) was applied consistently across experiments. Input and output texts were tokenized into integer sequences, padded with special tokens, and truncated to a maximum length of 512 tokens.

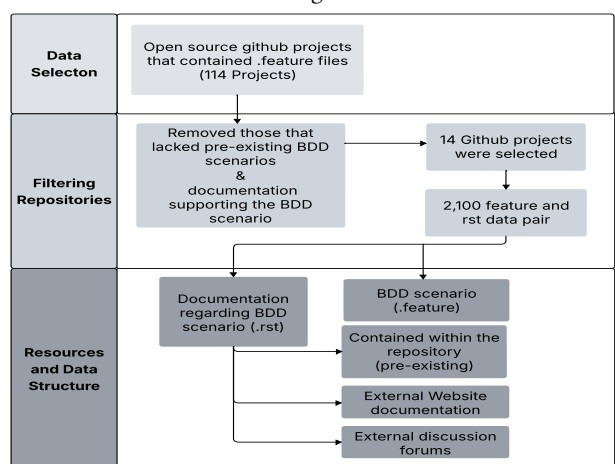

**Figure 5: Data collection process**

| Training Arguments | codet5p-220m | codet5p-770m |
|---|---|---|
| num-train-epoch | 15 | 12 |
| train-batch-size | 2 | 4 |
| eval-batch-size | 2 | 4 |
| warmup-steps | 10 | 10 |
| weight-decay | 0.01 | 0.01 |
| optim | adamw-torch | adamw-torch |
| dataloader-num-workers | N/A | 32 |
| bf16 | N/A | True |

**Table 1: Fine-tuning hyperparameters for the CodeT5p models**

**Context Management Strategies** Truncation was handled directly by the Hugging Face Tokenizer [3], which truncates token sequences to a specified maximum length. For the chunking strategy, we employed the RecursiveCharacterTextSplitter from the LangChain library [1], configured to segment text based on the target model's tokenizer. Abstractive summarization was performed using a pre-trained t5-base model [2] to generate a condensed summary of any text exceeding the context window.

## Part II: Prompt Engineering

**Models** We evaluated three model families for RAG few-shot prompting: Meta-Llama-3.1-8B [26], CodeT5+ [34], and DeepSeek-Coder [16]. Prompt-based generation was evaluated using the same metrics as the fine-tuning experiments—Exact match, BLEU, and token-based F1 score—allowing direct comparison between two approaches.

**Computational Environment** Full-scale experiments were executed on Kaggle using dual NVIDIA T4 GPUs.

**Dataset** We used the same curated dataset as in fine-tuning to ensure a fair comparison. The dataset was split at the individual pair level into a knowledge base (80%) and a validation set (20%),

**Role definition**
You are an expert Quality Assurance engineer specializing in writing Behavior-driven Development (BDD) scenarios in Gherkin syntax.

**INPUT**
Example 1:
```
{   "input Record": "{example_record_1}",
"BDD Scenario": "{example_scenario_1}" }
... (up to N retrieved examples are inserted here)
```

Task:
```
{                      "input          Record":
"{the_new_record_to_be_translated}",
"BDD Scenario": "" }
```

**Task instruction**
1) Translate the given "input Record" into a compliant and accurate "BDD Scenario".
2) Use the provided examples to follow the correct format and style.
3) After completing the BDD Scenario, stop and do not generate additional text.

**Figure 6: Prompt template for BDD scenario generation. Text in curly braces ({}) represents placeholders for dynamically filled content. The number of few-shot examples ($N$) is a variable in the experiments, typically set to 1 or 3.**

following standard retrieval-based evaluation protocols [21]. The knowledge base was used exclusively for retrieval, while the validation set was held out for evaluation. A retrieval pipeline was constructed using ChromaDB, an open input vector database [9]. The pipeline stored sentence-level embeddings with a dimensionality of 1,024, generated by the BAAI/bge-large-en-v1.5 sentence-transformer model. For each query, the system retrieved the top-K most similar scenarios/records from the knowledge base, where K was a hyperparameter set to 1 or 3 for the experiments.

## 6 Results and Analysis

This section reports the experimental results, structured around three research questions:

**RQ1: How do context management strategies affect model performance under a fixed 512-token context window?** This investigates the impact of different context management strategies—such as truncation, and chunking—on generation quality when using fine-tuning under strict context-length constraints.

**RQ2: What is the impact of task direction and instructional prefixing on generation quality?** This examines the effects of task direction (record, scenario) versus (scenario, record) and the use of instructional prefixes on model behavior and output quality.

**RQ3: How effective is fine-tuning compared to RAG few-shot prompting for record and BDD scenario generation?** This compares training-time adaptation through fine-tuning and also prompt engineering to determine which approach yields more accurate outputs.

| MethodStrategy | Exact Match | BLEU | F1-score |
|---|---|---|---|
| Unprefixed-Truncating | 0.0024 | 0.5245 | 0.6758 |
| Prefixed-Truncating | **0.0452** | **0.6123** | **0.7437** |
| Unprefixed-Summarizing | 0.0214 | 0.4429 | 0.6491 |
| Prefixed-Summarizing | 0.0238 | 0.4221 | 0.6423 |
| Unprefixed-TruncatingChunking | 0.0039 | 0.4591 | 0.6411 |
| Prefixed-TruncatingChunking | 0.0078 | 0.5363 | 0.6767 |
| Unprefixed-SummarizingChunking | 0.0020 | 0.4316 | 0.6307 |
| Prefixed-SummarizingChunking | 0.0039 | 0.5030 | 0.6804 |

Table 2: Fine-tuning (CodeT5p-220m): BDD scenario generation task

| MethodStrategy | Exact Match | BLEU | F1-score |
|---|---|---|---|
| Unprefixed-Truncating | 0.6024 | 0.8324 | 0.8788 |
| Prefixed-Truncating | 0.6048 | 0.8286 | 0.8805 |
| Unprefixed-Summarizing | 0.5571 | 0.7771 | 0.8502 |
| Prefixed-Summarizing | 0.6381 | 0.8224 | 0.8624 |
| Unprefixed-TruncatingChunking | 0.3198 | 0.4758 | 0.6443 |
| Prefixed-TruncatingChunking | **0.6816** | 0.8532 | **0.8927** |
| Unprefixed-SummarizingChunking | 0.2749 | 0.4305 | 0.6194 |
| Prefixed-SummarizingChunking | **0.6816** | **0.8574** | 0.8897 |

Table 4: Fine-tuning results (CodeT5p-220m): Record generation task

| MethodStrategy | Exact Match | BLEU | F1-score |
|---|---|---|---|
| Unprefixed-Truncating | 0.0476 | 0.6423 | 0.7472 |
| Prefixed-Truncating | 0.0714 | **0.6685** | **0.7697** |
| Unprefixed-Summarizing | **0.0762** | 0.5717 | 0.7282 |
| Prefixed-Summarizing | **0.0762** | 0.5864 | 0.7359 |
| Unprefixed-TruncatingChunking | 0.0705 | 0.5487 | 0.7018 |
| Prefixed-TruncatingChunking | 0.0744 | 0.6160 | 0.7436 |
| Unprefixed-SummarizingChunking | 0.0391 | 0.5350 | 0.6749 |
| Prefixed-SummarizingChunking | 0.0431 | 0.6050 | 0.7209 |

Table 3: Fine-tuning (CodeT5p-770m): BDD scenario generation task

| MethodStrategy | Exact Match | BLEU | F1-score |
|---|---|---|---|
| Unprefixed-Truncating | 0.7595 | **0.9394** | **0.9549** |
| Prefixed-Truncating | 0.7548 | 0.9276 | 0.9493 |
| Unprefixed-Summarizing | 0.7857 | 0.8974 | 0.9468 |
| Prefixed-Summarizing | **0.8119** | 0.9129 | 0.9516 |
| Unprefixed-TruncatingChunking | 0.4474 | 0.5699 | 0.7172 |
| Prefixed-TruncatingChunking | 0.7419 | 0.9238 | 0.9418 |
| Unprefixed-SummarizingChunking | 0.4067 | 0.5434 | 0.7010 |
| Prefixed-SummarizingChunking | 0.7027 | 0.9070 | 0.9360 |

Table 5: Fine-tuning results (CodeT5p-770m): Record generation task

## 6.1 RQ1

In this RQ, we study truncation, abstractive summarization, and one-to-many chunking, each applied in both *prefixed* and *unprefixed* input settings. These strategies are evaluated across two model scales, CodeT5p-220M and CodeT5p-770M, and for both generation directions (record-to-scenario and scenario-to-record). Tables 2, 4, 3, and 5 report the results.

**Results.** For the 220M model in the scenario generation task (Table 2), prefixed truncation yields the strongest results, while summarization can improve Exact Match in some unprefixed settings but typically reduces BLEU. In particular, *prefixed truncation* achieves the best Exact Match (0.0452) and BLEU (0.6123). Our inspection shows that summarization often removes or rewrites fine-grained details required for exact Gherkin reproduction (e.g., step parameters, ordering, and data-table rows). As a result, the generated scenarios are often semantically plausible but structurally mismatched. Chunking is not beneficial at this scale for scenario generation. Chunked outputs frequently lose global coherence across steps (e.g., repeating early steps, skipping later Then/And clauses, or producing incomplete tables), which prevents the model from producing a single unified scenario.

For the 220M model in the record generation task (Table 4), chunking helps *only when combined with prefixing*, and the improvement is modest but consistent on Exact Match and BLEU. Relative to the unprefixed truncation baseline (Exact Match 0.6024, BLEU 0.8324, F1 0.8788), *prefixed truncation + chunking* increases Exact Match to 0.6816 (+0.0792), BLEU to 0.8532 (+0.0208), and F1 to 0.8927 (+0.0139). A similar pattern holds for *prefixed summarization+chunking*. This setting more often preserves the record "skeleton" (Feature/Background + bullet checklist) while still producing later items instead of stopping early. In contrast, chunking without prefixing degrades performance (e.g., BLEU drops from

0.8324 to 0.4758 under unprefixed truncation + chunking), indicating that explicit task/continuation signaling is necessary for chunked training instances to be learnable.

Our qualitative investigation for the 220M model in the scenario generation task shows that prefixing is critical for task comprehension. Without the generate test scenario: instruction, the model frequently misunderstands the objective and defaults to generating contextually related but structurally incorrect artifacts, such as code snippets or documentation excerpts instead of the required Gherkin format. While this output is contextually relevant to the input's topic, it fails to adhere to the task's structural constraints. While prefixing corrects the task, chunking is not beneficial at this scale for scenario generation. Chunked outputs often lose global coherence across steps (e.g., repeating early steps or producing incomplete tables), which prevents the model from producing a single, unified scenario. This effect is consistent with the results in Table 2, where prefixed truncation yields the strongest performance.

For the 770M model, the relative behavior shifts. In the record generation task (Table 5), the strongest results are achieved by non-chunked strategies. Our inspection suggests the 770M model can maintain the global record structure in a single pass (Feature/Background + checklist) without relying on chunk-level continuation cues. Chunking reduces quality on BLEU and F1 (e.g., prefixed truncation + chunking: Exact Match 0.7419, BLEU 0.9238, F1 0.9418), suggesting that for larger models the one-to-many transformation is not necessary to achieve high-fidelity record outputs and may introduce avoidable inconsistency. Figure 3 illustrates a representative codet5p-770m output for record generation, showing that the model can preserve the record skeleton (Feature/Background+checklist) with minimal formatting noise.

In the scenario generation task with the 770M model (Table 3), truncation remains strong and prefixing provides a modest improvement: Exact Match increases from 0.0476 (unprefixed truncation) to 0.0714 (prefixed truncation), a gain of 0.0238, and BLEU rises from

0.6423 to 0.6685 (+0.0262). Summarization does not outperform truncation on BLEU in this setting. Chunking does not improve BLEU and F1 (e.g., prefixed truncation + chunking: Exact Match 0.0744, BLEU 0.6160, F1 0.7436), despite a small increase in Exact Match relative to prefixed truncation.

Our qualitative investigation for the 770M model shows that its increased capacity allows it to handle long-range dependencies and complex Gherkin structures in a single pass. In non-chunked settings, the model successfully maps individual record items to detailed, syntactically correct scenarios, showing that larger models can manage the one-to-many transformation without chunking. This effect is consistent with the results in Table 3, where chunking appears to be counterproductive because it breaks the input apart and makes it harder for the model to produce a unified output.

**Summary.** Truncation is a strong and stable baseline, particularly for scenario generation. Summarization is generally less effective than truncation for scenario generation on BLEU, although it can yield slightly higher Exact Match at 770M. Chunking is highly sensitive to both task direction and prefixing: it can yield small but measurable gains for 220M record generation when paired with explicit prefixing (up to +0.0792 Exact Match and +0.0208 BLEU over the unprefixed truncation baseline), but it consistently harms performance without prefixing and does not improve BLEU and F1 for the 770M model. These results show that context management choices must be reported jointly with task direction, prefixing, and model size, as their effects are not interchangeable across settings.

## 6.2 RQ2

This RQ investigates (i) how task direction affects generation quality and (ii) whether explicit prefixing improves performance.

**Results.** Tables 2, 4, 3, and 5 show that task direction has a substantial impact on generation quality. The *record* generation task consistently outperforms the reverse direction across all metrics, achieving high F1 scores (up to 0.9549), Exact Match (up to 0.8119), and BLEU scores (up to 0.9394). In contrast, the *scenario* generation task yields much lower Exact Match (up to 0.0762), despite moderate BLEU and F1 values (up to 0.6685 BLEU and 0.7697 F1-score), indicating difficulty in generating fully correct structured outputs. This gap is expected because Exact Match is brittle for Gherkin: minor differences in whitespace, keyword casing, step ordering, or data-table formatting count as mismatches even when the scenario is semantically correct.

Instructional prefixing has a limited and task-dependent effect. For the scenario generation task, prefixing improves performance across all metrics, yielding higher F1, Exact Match, and BLEU scores. However, for the record generation task, prefixing has mixed effects, suggesting that additional instructions may introduce unnecessary constraints when reconstructing free-form text.

**Summary.** Task direction is the dominant factor influencing generation quality, with record generation task substantially outperforming scenario generation. Instructional prefixing is beneficial primarily for structured output generation, while offering little or no advantage for less constrained generation tasks.

## 6.3 RQ3

This RQ examines whether RAG few-shot prompting can match, or meaningfully approximate, the performance of fine-tuning for record and BDD scenario generation tasks.

**Prompt engineering (RAG few-shot).** Table 6 reports the performance of RAG few-shot prompting using either one retrieved example (1-shot) or three retrieved examples (3-shot). The RAG in this study was set up for dynamic example retrieval rather than knowledge retrieval. For any given task, the RAG would search the knowledge base for the top-K source texts (records or scenarios) that were most semantically similar to the input. Once found, the complete (record, scenario) pairs were pulled and used as few-shot demonstrations in the prompt to show the model the desired output format. With `llama-3.1-8b`, RAG achieves an F1-score of 0.560 with 1-shot prompting and improves to 0.612 with 3-shot prompting, while BLEU increases from 0.2669 to 0.3474. For scenario generation task, F1-scores reach 0.413 with 1-shot prompting and 0.378 with 3-shot prompting, with BLEU scores remaining low at 0.1049 and 0.0953, respectively. Across all configurations, Exact Match remains at 0.0, indicating that RAG-based prompting fails to generate outputs that exactly conform to the required BDD format. Our qualitative inspection revealed that while many RAG outputs were partially correct, they frequently contained minor termination or formatting errors—such as trailing symbols, repeated prompt instructions, or a failure to stop correctly—which precluded an Exact Match success. Figure 2 shows an example output from `llama-3.1-8b` using 3-shot prompting; compared to the ground truth, it includes prompt artifacts (e.g., trailing braces such as }) and does not consistently stop at the intended endpoint, which prevents exact string matches. Overall, increasing the number of retrieved examples yields only marginal improvements, suggesting limited benefit from additional in-context demonstrations.

**Comparison with fine-tuning.** We compare RAG against the best-performing fine-tuned configuration per task from RQ1 and RQ2 reported in Tables 6: for scenario generation, CodeT5p-770m with prefixed-truncating (best BLEU/F1), and for record generation, CodeT5p-770m with unprefixed-truncating (best BLEU/F1). We additionally report the highest Exact Match configuration for record generation (prefixed-summarizing) to reflect strict string-level matches. As shown in Table 6, fine-tuning substantially outperforms the best RAG configuration on metrics that require strict structural and surface-form similarity. For scenario generation, the best fine-tuned configuration (CodeT5p-770m, prefixed-truncating) achieves Exact Match 0.0714, BLEU 0.6685, and F1 0.7697, compared to Exact Match 0.000 and BLEU 0.1049 for 1-shot RAG prompting with `llama-3.1-8b` (F1 0.4134). For record generation, the best fine-tuned configuration by BLEU/F1 (CodeT5p-770m, unprefixed-truncating) achieves Exact Match 0.7595, BLEU 0.9394, and F1 0.9549, while the highest-Exact-Match configuration (prefixed-summarizing) achieves Exact Match 0.8119. In contrast, the best RAG configuration attains F1 0.6120, with Exact Match 0.000 and BLEU 0.3474.

Our qualitative investigation for RAG showed the reasons it did not work well was due to to the models' tendency to hallucinate prompt-related structural elements and conversational filler rather than producing a clean BDD artifact. Specifically, outputs across all models frequently included unnecessary closing brackets (e.g., " or

| Generation Task | Approach | Model | Exact Match | BLEU | F1-score | Human Success Rate(%) | LLM Judge(GPT-oss-120b) | LLM judge(Qwen-3-32b) |
|---|---|---|---|---|---|---|---|---|
| Scenario generation | Fine-tuning (prefixed-summarizing) | codet5p-770m | 0.0762 | 0.5864 | 0.7359 | 100% | 30% | 40% |
| Record generation | Fine-tuning (prefixed-summarizing) | codet5p-770m | **0.8119** | **0.9129** | **0.9516** | 100% | 90% | 100% |
| Scenario generation | RAG few-shot (shots = 1) | llama-3.1-8b | 0.0000 | 0.1049 | 0.4134 | 10% | 0% | 10% |
| Record generation | RAG few-shot (shots = 1) | llama-3.1-8b | 0.0000 | 0.2669 | 0.5601 | 90% | 0% | 20% |
| Scenario generation | RAG few-shot (shots = 3) | llama-3.1-8b | 0.0000 | 0.0953 | 0.3780 | 20% | 10% | 30% |
| Record generation | RAG few-shot (shots = 3) | llama-3.1-8b | 0.0000 | 0.3474 | 0.6120 | 60% | 0% | 30% |
| Scenario generation | RAG few-shot (shots = 1) | codet5p-770m | 0.0000 | 0.0781 | 0.3214 | 30% | 0% | 0% |
| Record generation | RAG few-shot (shots = 1) | codet5p-770m | 0.0000 | 0.2083 | 0.4852 | 70% | 0% | 10% |
| Scenario generation | RAG few-shot (shots = 3) | codet5p-770m | 0.0000 | 0.0386 | 0.2173 | 0% | 0% | 0% |
| Record generation | RAG few-shot (shots = 3) | codet5p-770m | 0.0000 | 0.0632 | 0.3123 | 0% | 0% | 10% |
| Scenario generation | RAG few-shot (shots = 1) | codet5p-220m | 0.0000 | 0.0850 | 0.3325 | 10% | 10% | 10% |
| Record generation | RAG few-shot (shots = 1) | codet5p-220m | 0.0000 | 0.2275 | 0.5612 | 50% | 0% | 0% |
| Scenario generation | RAG few-shot (shots = 3) | codet5p-220m | 0.0000 | 0.0781 | 0.3463 | 20% | 0% | 20% |
| Record generation | RAG few-shot (shots = 3) | codet5p-220m | 0.0000 | 0.1753 | 0.5252 | 50% | 0% | 0% |
| Scenario generation | RAG few-shot (shots = 1) | deepseek-coder-33b | 0.0000 | 0.0569 | 0.2539 | 10% | 0% | 0% |
| Record generation | RAG few-shot (shots = 1) | deepseek-coder-33b | 0.0000 | 0.0559 | 0.2654 | 0% | 0% | 0% |
| Scenario generation | RAG few-shot (shots = 3) | deepseek-coder-33b | 0.0000 | 0.0492 | 0.2723 | 0% | 0% | 0% |
| Record generation | RAG few-shot (shots = 3) | deepseek-coder-33b | 0.0000 | 0.0563 | 0.2338 | 0% | 0% | 0% |

**Table 6: Comparison of fine-tuning vs. RAG prompt engineering**

}), repeated the instructional prompt, or failed to stop after the task was complete, which explains why the Exact Match score remained at 0.00.

**Human Success Rate.** We also conducted a manual human evaluation of 10 generated outputs for each approach (18 configurations) in Tables 6. Each output was assessed for its correspondence to the ground-truth reference using a four-point rubric: (1) (Total Failure): The output is irrelevant, nonsensical, or fails to address the prompt. (2) (Partially Correct): The output contains significant structural or semantic errors but captures some key elements. (3) (Mostly Correct): The output is structurally sound and semantically correct, with only minor errors or omissions. (4) (Perfect): The output perfectly matches the expected structure and meaning. A success threshold was defined as a score of 3 ("Mostly Correct") or 4 ("Perfect") on the four-point rubric. The "Human Success rate (%) reported in Table 6 was then calculated as the percentage of the 10 evaluated samples for each approach that met this threshold. Our human evaluation shows that many generations remain practically usable even when small formatting differences prevent an exact string match. This indicates that strict automatic metrics can understate the actual functional correctness for both scenario and record generation.

**LLM Judge.** We also used LLM judges to provide an automated layer of evaluation. The assessment was conducted using two models: GPT-oss-120b and Qwen-3-32b, both accessed via the Cerebras API. We used the same 10 samples per approach from the human evaluation. A prompt was sent to both models for each sample, instructing them to compare the generated output against the expected (ground-truth) output using the same four-point rubric as the human evaluation. To enable a direct comparison, we applied the same success threshold to both judges: a sample was considered successful if it received a score of 3 or 4. This allows us to calculate an LLM Judge *success rate* (%), defined as the fraction of samples scored as 3 or 4 (out of 10), which we report in Table 6, which can be directly compared to the manual Human success rate.

The assessment reveals that LLM judges tend to be stricter regarding surface-form and prompt artifacts than human reviewers. This behavior appears to align with recent findings suggesting that increased reasoning overhead in certain models can impact reliability when navigating complex instructions [19]. Additionally,

the judges exhibited tendencies consistent with known scoring biases and sensitivity to structural perturbations, which can lead to fluctuations in scoring consistency [31].

**Summary.** While RAG few-shot prompting can capture partial lexical overlap, as reflected by moderate F1-scores, it fails to generate syntactically complete and structurally valid BDD outputs. In contrast, fine-tuning consistently produces outputs with high Exact Match and BLEU scores, demonstrating superior structural and lexical fidelity. These findings indicate that fine-tuning remains the more effective approach for BDD generation when strict output correctness is required.

## 7 Limitations

*Dataset composition.* Although the dataset is curated from 14 real-world GitHub repositories, the documentation and BDD styles may not represent all domains. The dataset may also contain patterns that favor one adaptation approach (e.g., fine-tuning) over another. To mitigate noise from open-source artifacts, we filtered for completeness (records with matching scenarios) and decomposed complex nested scenarios into atomic parent scenarios. Also, a larger and more diverse dataset would increase statistical power and confidence in the observed differences.

*Model specificity.* We fine-tuned a code-centric LLM at two different model sizes and applied prompt engineering across multiple models, including larger frontier models. Performance trends may still differ under other model choices, architectures, dataset scales, or stronger proprietary/frontier models, and different configurations could yield different results.

## 8 Conclusion

This paper studied bidirectional generation between records and BDD scenarios and compared fine-tuning of CodeT5+ models with RAG few-shot prompting using Meta-Llama-3.1-8B, CodeT5+ models, and DeepSeek-Coder. Across 2,100 aligned pairs, results show that task direction and context management choices substantially affect performance, and that instructional prefixing mainly benefits tasks requiring strict structural outputs. Overall, fine-tuning produces structurally reliable BDD artifacts.

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
