# OpenReview forum: "Generating Behavior-Driven Development (BDD) Artifacts"
_ACM.org/AIWare/2026/Conference — Submitted to AIware 2026_

### Official Review · Reviewer_fkzv · 2026-03-08

**Rating:** 2
**Confidence:** 4

**Review:**

**Strengths**

* The curated dataset of 2100 (record, scenario) pairs from GitHub repos appears to require substantial curation effort and can be very valuable for future work in this space.
* The bidirectional formulation is reasonable and interesting, which helps reveal the difficulty of these two directions.
* The paper is easy to follow.


**Weaknesses**

* The evaluated models feel limited and somewhat dated for this task. The fine-tune study was conducted on CodeT5+ models with a 512-token context window, which may be limited for such task. The task itself does not seem to require an encoder–decoder architecture. It’s therefore unclear the reported challenges, such as long-input handling, reflects the task itself versus the constraints of the chosen models.
* The evaluation relies mainly on Exact Match, BLEU and F1, which are lexical metrics and can be brittle. Even the paper itself notes that Exact Match is brittle for Gherkin, as minor formatting differences can count as mismatches even when outputs are semantically acceptable. Although the paper includes a small scale human evaluation and LLM judge, analyses are not central enough to fully offset the limitations of the main metrics.


**Questions / Comments for author**

* Do the authors consider conducting the study under more recent models, especially models with larger context windows?
* The current 80/20 split was performed at paired instance level. I am curious whether the authors considered a repo-level split as well to assess cross-repo generalization.
* Since the paper itself already notes that Exact Match is brittle, do the authors consider evaluation metrics that are more semantic-aware. I noticed that the paper includes a small scale evaluation with LLM judge. I was not fully convinced by the conclusion that the LLM judge is biased. My reading is that LLM judge may actually be a more suitable evaluation appraoch for this task, but that the current judge may not be robust enough for the evaluation. Could the authors clarify why they view the judge behavior mainly as bias?

I think this paper provides an empirical study but is somewhat limited. The task is interesting, the dataset is valuable, and the paper makes an attempt at a controlled comparison. However, the conclusions are based on relatively constrained model choices, a somewhat weak non-fine-tuning baseline, and evaluation metrics that may not align well with semantic generation quality.

**Summary:**

This paper studies bidirectional generation between semi-structured records and structured BDD scenarios using LLMs. It compares fine-tuning CodeT5+ encoder–decoder models against RAG-based few-shot prompting on a curated dataset of 2100 aligned pairs from 14 GitHub repositories, and also studies truncation, summarization, and chunking for long inputs. The main result is that, under the authors’ setup, fine-tuning outperforms the chosen RAG baselines, especially for record generation, while scenario generation remains substantially harder.

---

> ### Author Response · Authors · 2026-03-19
>
> ```markdown id="rzmkjh"
> We thank the reviewer for the careful and constructive feedback. We appreciate the thoughtful questions. Below we respond point by point, combining overlapping comments where appropriate.
>
> **Model choice, context window limits, and architecture selection**
>
> **Response:**
> We thank the reviewer for this helpful comment and agree that this aspect of the manuscript requires clearer justification.
> We selected CodeT5+ because the task is inherently sequence-to-sequence, involving the transformation of semi-structured records into structured BDD scenarios and vice versa. Encoder–decoder models are well suited to this type of generation task. We agree that decoder-only architectures are highly relevant, especially given current trends in code and language models. While decoder-only models were included in our RAG experiments, they were not explored in the fine-tuning setting. We will revise the manuscript to better justify the choice of encoder–decoder models and to discuss decoder-only fine-tuning more explicitly as an important direction for future work.
> Regarding context length, the use of CodeT5+ with a 512-token limit was intentional. One objective of the study was to investigate how context-management strategies such as truncation, summarization, and chunking perform under constrained input conditions. This allowed us to examine these trade-offs directly. However, we agree that the long-input limitations observed in our experiments should be understood in the context of the selected CodeT5+ models and their 512-token limit, rather than as inherent properties of the task itself. We will make this distinction more explicit in the revised manuscript.
> We also agree that evaluating newer models with larger context windows is an important next step. Such models may reduce information loss by enabling full technical records and multi-step scenarios to be processed in a single pass, without relying on truncation or summarization. They may also enable a more direct comparison between large general-purpose models and smaller domain-specific fine-tuned models for BDD automation.
>
> **Repository-level split and cross-repo generalization**
>
> **Response:**
> We appreciate this observation and agree that a repository-level split would provide a stronger evaluation of cross-repository generalization.
> In this study, we used an instance-level split because the primary goal was to compare fine-tuning and RAG under a shared, diverse data distribution. Since the dataset was curated from 14 repositories, this approach exposed the models to a broader range of Gherkin styles and record formats during training while preserving diversity across the training and validation sets. We will clarify this choice in the manuscript, and identify repository-level evaluation as an important direction for future work.
>
> **Lexical metrics, semantic-aware evaluation, and the LLM judge**
>
> **Response:**
> We thank the reviewer for this thoughtful point and agree that semantic-aware evaluation is a valuable complement for BDD tasks.
> Exact Match was included because it reflects the strict precision often required for executable scenarios, but it should not be interpreted in isolation. We will revise the manuscript to more clearly frame Exact Match, BLEU, and F1 as baseline lexical measures, and to better emphasize the complementary role of the human and LLM-based evaluations.
> Regarding the LLM judge specifically, our intent was to highlight that the current judge did not reliably distinguish correct local segments from surrounding noisy output. In several cases, human evaluation recognized partially correct content and assigned higher scores accordingly, while the LLM judge assigned uniformly low scores. We agree with the reviewer that “lack of robustness” is a more precise characterization than “bias,” and we will revise the manuscript to reflect that distinction more carefully.
> ```

---

### Official Review · Reviewer_Y1BM · 2026-03-11

**Rating:** 2
**Confidence:** 3

**Review:**

## Strength
+ The paper studies a less explored bidirectional setting, record-to-scenario and scenario-to-record generation, rather than only forward BDD scenario generation.
+ The problem is practically relevant, because BDD projects often contain both executable scenarios and less-structured supporting records, and linking the two is labour-intensive.
+ Presentation: The paper is generally easy to follow, with a clear problem statement, a concrete running example, and results organised around explicit research questions.

## Weakness
- The main contribution is strongest in the task framing and dataset curation, while the modelling techniques themselves are more incremental.
- Some of the paper’s broader claims, especially around fine-tuning versus RAG, feel somewhat stronger than what the evidence can fully support.
- The comparison between fine-tuning and RAG is not entirely balanced, and the split design leaves room for alternative interpretations.
- The paper gives a solid overview of the setup, but several details would still need to be added for full reproducibility.

## Detailed Comments

### Novelty

The study has novelty in its bidirectional framing and in the creation of a dataset of aligned BDD records and scenarios drawn from real GitHub repositories. That is a useful step beyond much of the prior work, which tends to focus mainly on one-way scenario generation from requirements-like inputs. In that sense, they offer a meaningful extension of the problem space.

However, I think the novelty claim would be stronger if it were framed a bit more precisely. The individual techniques used here, such as fine-tuning, retrieval-augmented prompting, truncation, summarization, and chunking, are all established methods. Because of that, the main originality seems to lie less in proposing a new method and more in combining existing methods within a new task setting and evaluating them carefully on a curated dataset. That is still a valid and publishable contribution, but I would encourage the authors to position it in those terms.

I also think the related-work discussion could be a bit more measured. The study is likely novel in its exact bidirectional setup and evaluation design, but some of the surrounding literature on BDD generation and LLM adaptation in software engineering is broader than the current framing suggests. Narrowing the claim slightly would make the contribution feel more credible rather than less.

### Significance

This is a practically meaningful problem. In real BDD workflows, teams often maintain executable scenarios alongside less structured artefacts such as issue reports, notes, or documentation, and mapping between those artefacts can be tedious and error-prone.

The dataset also strengthens the study’s practical value. Collecting 2,100 aligned pairs from 14 repositories is not trivial, and this gives the study a more realistic basis than many small or synthetic evaluations.

My concern is mainly about how broadly the conclusions are phrased. The results do support the claim that the fine-tuned setups in this study outperform the RAG setups in this study, especially under strict structural correctness criteria. However, I am less convinced that the evidence supports a broader takeaway that fine-tuning is generally the better adaptation strategy for this task family. The study would be stronger if it stated the conclusion a bit more narrowly and tied it more explicitly to the specific experimental conditions evaluated here.

I also think the practical significance would increase considerably if the evaluation connected more directly to downstream BDD use, for example, by checking executability or tool compatibility of generated scenarios. At present, the results are useful, but still somewhat indirect from a practitioner’s perspective.

### Soundness

This is the area where I had the most concerns, although I think many of them are fixable through clarification and reframing.

The main issue is that the fine-tuning versus RAG comparison does not feel fully balanced. Fine-tuning benefits from several task-specific design choices, including supervision on the target dataset, prefixing, summarisation, truncation, and chunking. By contrast, the RAG setup appears more limited, and the authors explicitly note that comparable length-handling strategies were not applied there. Since the qualitative analysis suggests that many RAG errors are tied to formatting, stopping, or prompt artefacts, it becomes difficult to tell whether they are identifying a limitation of RAG as a paradigm or mainly a limitation of the particular RAG implementation used here. I do think the experimental result is still informative, but the interpretation should be a bit more cautious.

A second concern is the split design. Because the data come from only 14 repositories and are split at the pair level rather than by repository, there is a possibility that models are benefiting from repository-specific style or template regularities that appear in both training and validation. This may be especially relevant for record generation, where preserving structural patterns seems important. I am not saying this invalidates the results, but it does leave open an alternative explanation that the authors should acknowledge more directly. A cross-repository evaluation would make the generalisation claim much stronger.

I also found the treatment of chunking somewhat difficult to interpret. Chunking is introduced as a context-management strategy, but it also changes the effective training distribution and introduces continuation-related prompt signals. As a result, it is hard to isolate exactly what part of the performance difference is due to chunking itself. This does not make the experiment unhelpful, but it does mean the conclusions about why chunking helps should be presented with more care.

Finally, the additional evaluation components, especially the human review, feel too limited to strongly arbitrate between configurations. Reviewing only 10 outputs per approach gives a useful qualitative snapshot, but it is a small sample for supporting comparative claims across many settings. More detail on the evaluation protocol, raters, and agreement would make this part much more convincing. Similarly, the LLM-judge setup would benefit from fuller specification.

### Presentation

Overall, I found the manuscript readable and fairly easy to follow. The motivation is clear, the workflow is understandable, and the running example helps ground the task. The organisation by research question also works well and makes the experimental section manageable.

However, the draft still feels a little unfinished in places. The most important issue is the inconsistency between the RQ3 prose and Table 6. In one place, the text says the comparison uses one fine-tuning configuration for scenario generation and another for record generation, while Table 6 appears to report a different configuration for both. This is not just a cosmetic issue, because it affects the credibility of the study’s main comparative claim. I would strongly encourage the authors to reconcile this carefully.

I also think some of the language could be softened and made more precise. In a few places, the authors seem to present the RAG results in a more categorical way than the evidence fully supports. Since they already include nuanced findings, including partial success under some conditions, the writing would benefit from reflecting that nuance consistently.

There are also several template remnants and small drafting errors. None of these is fatal, but together they give the impression that the manuscript is still one revision away from being submission-ready.

### Transparency/Reproducibility

The authors do a good job of reporting the overall design of the study. Readers are told which model families are used, what the main training settings are, what retrieval components are included, and which evaluation metrics are applied. That gives a solid foundation for understanding the work.

My concern is mainly that some of the finer-grained details needed for reproduction are still missing. For example, it is not yet fully clear whether the dataset, preprocessing scripts, prompt templates, and evaluation artefacts will be released. Since the dataset is one of the key contributions, that matters quite a bit. The same applies to details of chunking, summarisation, retrieval prompt construction, and the LLM-judge protocol.

The human evaluation section would also benefit from more explicit reporting. Information such as the number of raters, whether ratings were blind, how disagreements were handled, and whether agreement was measured would help readers interpret those results with more confidence.

So overall, I would describe this aspect as promising but not yet complete. The manuscript is already transparent at a high level, but it would benefit from more implementation and protocol detail to fully support reproducibility.

## Minor Comments

1. Page 2: “Sections 7 and 8 discuss limitations and and future research.” The duplicated and should be removed.
2. Page 7 versus Table 6 on page 8: The prose describing which fine-tuning configuration is used in the comparison does not appear to match the table.
3. Page 7: “the reasons it did not work well was due to to...” contains both a duplicated to and a subject-verb problem.

**Summary:**

This paper studies whether large language models can automatically translate back and forth between informal software records, such as documentation or issue text, and structured BDD scenarios. Using a curated dataset of 2,100 aligned record-scenario pairs from 14 GitHub repositories, it compares fine-tuned CodeT5+ models against retrieval-augmented few-shot prompting and finds that fine-tuning is clearly stronger, especially for producing structurally correct outputs, while performance also depends on task direction, context handling, and whether explicit instructional prefixes are used.

---

> ### Author Response · Authors · 2026-03-20
>
> ```markdown id="lbwlkb"
> We thank the reviewer for the careful and constructive feedback. Below we respond to the questions.
>
> **Novelty**
> **Response:**
> We agree. The primary contribution of this work is empirical rather than algorithmic. More specifically, the novelty lies in the bidirectional BDD framing, the curated dataset of 2,100 aligned pairs from 14 real GitHub repositories, and the controlled comparison of adaptation strategies, architectures, and context-management methods in this setting. We will revise the manuscript to position the work more clearly in those terms and to broaden the Related Work discussion to better situate it within prior research on BDD generation and LLM adaptation in software engineering.
>
> **Significance**
> **Response:**
> In the revision, we will narrow our conclusions to the specific models, configurations, and evaluation conditions studied here, rather than implying a broader claim that fine-tuning is generally superior for this task family. We also agree that downstream validation would strengthen the practical significance of the work, and we will identify parser-based or execution-based evaluation in a BDD framework as an important direction for future work.
>
> **Soundness**
> **Comment:**
> The fine-tuning versus RAG comparison
> **Response:**
> The RAG setup was intentionally evaluated in a relatively standard few-shot setting rather than as a highly optimized pipeline, but this does limit the scope of the comparison. We will revise the manuscript to make clear that our findings apply to the specific RAG implementation tested here rather than to RAG as a general paradigm.
>
> **Comment:**
>  A repository-level split
> **Response:**
> In this study, we used an instance-level split because the primary goal was to compare fine-tuning and RAG under a shared, diverse data distribution. Since the dataset was curated from 14 repositories, this approach exposed the models to a broader range of Gherkin styles and record formats during training while preserving diversity across the training and validation sets. We will clarify this choice in the manuscript, and identify repository-level evaluation as an important direction for future work.
>
> **Comment:**
> The treatment of chunking
> **Response:**
> Chunking affects prompt structure, continuation signals, and the effective training distribution, so its benefits cannot be attributed to a single isolated factor. We will revise the manuscript to clarify that our results reflect the combined effect of the chunking strategy.
>
> **Comment:**
> The human and LLM-judge evaluations
> **Response:**
> We agree that the human and LLM-judge evaluations should not be interpreted as definitive arbiters between configurations. Their purpose was to serve as qualitative complements to the large-scale automated evaluation, and we will clarify this scope more explicitly in the revised manuscript.
> For the human evaluation, we selected a representative subset of 180 generated artifacts, with 10 samples drawn from each of the 18 experimental configurations. Two experts independently evaluated each sample by reviewing the original input together with the model-generated output and assigning scores using a standardized 4-point rubric. To enable direct comparison between manual and automated assessment, the same 180-sample subset was also evaluated by two LLM judges, GPT-OSS-120B and Qwen-3-32B.
> Inter-annotator agreement over the 180 binary correctness judgments yielded Cohen’s kappa values ranging from 0.00 to 1.00, with a mean of 0.46, and raw agreement ranging from 30% to 100%, with a mean of 77.8%. These results indicate moderate overall agreement, which is typical for subjective technical quality assessment tasks. Disagreements were resolved through consensus discussion, and the final agreed-upon judgments were used in the analysis.  We will revise the manuscript to describe this protocol in greater detail.
>
> **Presentation**
> **Response:**
> We agree that it is important and will correct the inconsistency so that the reported configurations are fully aligned throughout the manuscript.
>
> **Comment:**
> The presentation of the RAG results.
> **Response:**
> We agree. We will soften categorical statements about RAG and revise the discussion so that it more accurately reflects the task-dependent nuances observed in the results.
> We will also perform a careful proofreading pass to remove template remnants, duplicated words, and other drafting issues.
>
> **Transparency / Reproducibility**
> **Response:**
> We appreciate the reviewer’s concern regarding reproducibility. We have made the dataset, preprocessing scripts, prompt templates, and experiment artifacts publicly available here: https://github.com/Anonymous5420935/Project_Submission. In the revised manuscript, we will also add clearer implementation details for chunking, summarization, retrieval prompt construction, and the LLM-judge protocol to make reproduction easier.
> ```

---

> > ### Comment · Reviewer_Y1BM · 2026-03-20
> >
> > Thank you for the rebuttal. I appreciate the clarifications and the willingness to tighten the paper’s positioning. The responses on novelty, significance, presentation, and reproducibility address my concerns at a broad level, and the added detail on the human and LAG-judge evaluation makes that part of the paper much clearer.
> >
> > My main remaining concerns are still around soundness, and they are only partially addressed. On the fine-tuning versus RAG comparison, narrowing the claim to the specific RAG setup studied here is helpful, but the underlying comparison is still asymmetric, so I do not think the central takeaway is fully resolved. Similarly, I understand the rationale for the instance-level split, but this does not rule out the alternative explanation that some of the gains may reflect repository-specific regularities rather than stronger cross-project generalisation.
> >
> > The clarification on chunking is helpful, and I appreciate the commitment to revise the manuscript accordingly. Overall, the rebuttal improves the framing and transparency of the paper, but it does not fully resolve my main soundness concerns. For that reason, I am not changing my score, and my overall assessment remains Weak Reject.

---

### Official Review · Reviewer_gkNm · 2026-03-12

**Rating:** 4
**Confidence:** 3

**Review:**

Strengths

Relevant problem in software engineering. Automating BDD artifacts is a practical and useful problem for modern testing workflows.

Clear experimental scope. The paper studies a fairly comprehensive set of research questions, including the effect of context management strategies, task direction, instructional prefixing, and the comparison between fine-tuning and RAG-based prompting.

Moderate empirical contribution. While the techniques themselves are not new, the paper provides a controlled empirical comparison of adaptation strategies for BDD artifact generation using the same dataset and evaluation setup.

Qualitative analysis. The paper includes qualitative examples and error analysis, which help explain the observed quantitative results.

Weaknesses

1. Limited methodological novelty

The main techniques used in the study (fine-tuning, RAG prompting, prefixing strategies, and context truncation/summarization) are well-established. The contribution is primarily empirical rather than proposing a new modeling or algorithmic approach.

2. Model selection choices are not fully justified

The fine-tuning experiments focus on encoder–decoder models (CodeT5+), which are suitable for structured generation tasks. However, it would be useful to better justify this choice and discuss why decoder-only models were not explored in the fine-tuning setting, especially since many modern code models follow that architecture.

3. Evaluation focuses on surface metrics

The evaluation primarily relies on Exact Match, BLEU, and token-level F1. While these metrics are common, Exact Match can be particularly brittle for structured artifacts like Gherkin scenarios. Additional structure-aware evaluation or task-level validation could strengthen the conclusions. Though, great to see human success rate and LLM judge in Table-6

Suggestions for Improvement

1. Clarify the rationale for choosing encoder–decoder models for fine-tuning, and discuss how the results might differ with decoder-only architectures.

2. Consider including structure-aware evaluation metrics or automated validation of generated BDD artifacts.

3. Improve the presentation in some sections by reducing repetition and highlighting the key insights from the experiments more clearly.

**Summary:**

This paper studies the use of large language models for generating Behavior-Driven Development (BDD) artifacts, focusing on two directions: generating Gherkin scenarios from semi-structured records and regenerating records from scenarios. The authors curate a dataset of 2,100 aligned (record, scenario) pairs from 14 GitHub repositories and compare two adaptation strategies: fine-tuning CodeT5+ encoder–decoder models and RAG-based few-shot prompting with models including Llama-3.1-8B and DeepSeek-Coder.

The study investigates several factors affecting generation quality, including task direction, context management strategies (truncation, summarization, chunking), and instructional prefixing. Results show that fine-tuned models achieve substantially better structural correctness for BDD artifacts than RAG-based prompting under the evaluated settings.

Overall Assessment

This paper explores a practical application of LLMs for BDD artifact generation and provides a reasonably comprehensive empirical study of several factors affecting generation quality. While the methodological novelty is limited, the experimental comparison and analysis may still be useful for practitioners and researchers working on LLM-assisted software engineering tools.

---

> ### Author Response · Authors · 2026-03-19
>
> ```markdown
> We thank the reviewer for the constructive suggestions. We agree that these clarifications and revisions would strengthen the manuscript. Our responses are provided below.
>
> *Clarify the rationale for choosing encoder–decoder models for fine-tuning, and discuss how the results might differ with decoder-only architectures.*
>
> **Response:**
> We appreciate this suggestion. We selected CodeT5+ because the task is inherently sequence-to-sequence, involving the transformation of semi-structured records into structured BDD scenarios and vice versa. Encoder–decoder models are well suited to this type of generation task because they explicitly model the mapping from an input representation to a target output representation.
> We agree, however, that decoder-only architectures are also highly relevant, especially given current trends in code and language models. Their performance in this setting may differ because they rely on autoregressive generation within a single stream, rather than an explicit encoder–decoder structure. In practice, decoder-only models may benefit from larger scale and longer context windows, but they may also exhibit different trade-offs in structured transformation tasks such as record-to-scenario generation. While decoder-only models were included in our RAG experiments, they were not explored in the fine-tuning setting. We will revise the manuscript to better justify the choice of encoder–decoder models and to discuss decoder-only fine-tuning more explicitly as an important direction for future work.
>
> *Consider including structure-aware evaluation metrics or automated validation of generated BDD artifacts.*
>
> **Response:**
> We agree that structure-aware evaluation or automated validation would strengthen the study. In the current version, Exact Match, BLEU, and F1 were used as baseline lexical metrics, but we recognize that these measures do not fully capture structural correctness or practical usability for BDD artifacts such as Gherkin scenarios.
> For this reason, we also included human success rate and LLM-based evaluation as complementary analyses. In the revision, we will make this framing clearer and explicitly note that parser-based validation, executability checks, or other structure-aware evaluation methods would be valuable extensions for future work.
>
> *Improve the presentation in some sections by reducing repetition and highlighting the key insights from the experiments more clearly.*
>
> **Response:**
> We thank the reviewer for this suggestion and agree that the presentation can be improved. In the revision, we will reduce repetition, tighten the discussion in several sections, and revise the presentation of the results to highlight the main experimental insights more clearly.
> We will also make the discussion more precise in places where the current wording is broader or more categorical than the evidence supports, particularly in the interpretation of the comparative findings.
> ```

---

### Author Response · Authors · 2026-03-20

Dear Reviewers and Program Co-Chairs,

Thank you for your insightful and constructive feedback and for the opportunity to respond.

To support transparency and reproducibility, we have made the dataset, preprocessing scripts, prompt templates, and experiment artifacts publicly available in the following anonymous repository: https://github.com/Anonymous5420935/Project_Submission.